# System Dynamics Model for the Improvement Planning of School Building Conditions

**Suhyun Kang [1], Sangyong Kim [1], Seungho Kim [2] and Dongeun Lee [3],***

[1]  School of Architecture, Yeungnam University, 280 Daehak-ro, Gyeongsan-si, Gyeongbuk 38541, Korea; yp043422@ynu.ac.kr (S.K.); sangyong@yu.ac.kr (S.K.)

[2]  Department of Architecture, Yeungnam University College, 170 Hyeonchung-ro, Nam-gu, Daegu 42415, Korea; kimseungho@ync.ac.kr

[3]  School of Architecture & Civil and Architectural Engineering, Kyungpook National University, 80 Daehak-ro, Buk-gu, Daegu 41566, Korea

*  Correspondence: dolee@knu.ac.kr; Tel.: +82-53-950-7540

**Abstract:** As the number of aged infrastructures increases every year, a systematic and effective asset management strategy is required. One of the most common analysis methods for preparing an asset management strategy is life cycle cost analysis (LCCA). Most LCCA-related studies have focused on traffic and energy; however, few studies have focused on school buildings. Therefore, an approach should be developed to increase the investment efficiency for the performance improvement of school buildings. Planning and securing budgets for the performance improvement of school building is a complex task that involves various factors, such as current conditions, deterioration behavior and maintenance effect. Therefore, this study proposes a system dynamics (SD) model for the performance improvement of school buildings by using the SD method. In this study, an SD model is used to support efficient decision-making through policy effect analysis, from a macro-perspective, for the performance improvement of school buildings.

**Keywords:** school buildings; system dynamics; deterioration; rehabilitation; lifecycle cost analysis; budget allocation

---

## 1. Introduction

Recently, due to the rapid increase in deteriorated social infrastructures, the significance of long-term planning for sustainability and performance improvement has been noted. In the United States, the facility deterioration problem was noted in the 1980s, and in 2017, the condition grade of infrastructure was confirmed to be "D+" on average. In particular, according to the '2017 infrastructure report card', which was published by the American Society of Civil Engineers (ASCE), the required restoration cost is approximately KRW 1,120 trillion. In Japan, 63% of roads and bridges, 62% of river management facilities, and 58% of harbors and wharves in 2033 will have passed over 50 years of age after construction. Therefore, in major advanced countries, the rapid deterioration of social infrastructures has been caused by the lack of appropriate measures and investments, despite the increase in deteriorated social infrastructures [1]. Furthermore, because of climate change and the frequent occurrence of natural disasters (i.e., earthquakes and storms) worldwide, many human lives are lost during disasters such as the collapse of deteriorated bridges [2]. Therefore, the importance of life extension and performance improvement of existing deteriorated social infrastructures is stressed, to ensure the safety of people from these disasters and catastrophes.

Systematic and effective asset management strategies are required to solve these problems. One of the most common analysis methods for preparing an asset management strategy is life cycle cost

analysis (LCCA). Most LCCA-related studies have focused on traffic [3], pavement [4] and energy [5,6], however, few studies have focused on school buildings accommodating a population of over 50 million people on a daily basis.

In recent years, many studies have discussed the performance improvement of school buildings; however, the governments of various countries are experiencing difficulties in planning and financing, because of the lack of comprehensive data regarding school buildings [1]. Until recently, the maintenance of most school buildings was conducted using a breakdown maintenance method, instead of a preventive maintenance method. This method has led to the rapid deterioration of school buildings because the appropriate maintenance period was missed. Currently, the governments of advanced countries are hurriedly allocating excessive budgets; however, because executing these budgets within a financial year is an impossible task, the budget is customarily carried over to the following year, every year. This phenomenon is seemingly a result of short-term and emergency response, instead of investments based on mid-/long-term planning for the performance improvement of school buildings. Therefore, an approach should be developed to increase the investment efficiency for the performance improvement of school buildings. Planning and securing budgets for the performance improvement of social infrastructures is a complex task that involves various factors, such as current conditions, deterioration behavior, and maintenance effect [7]. Based on these complexities, it will be advantageous to predict the policy effect by using a simulation, to ensure that the policy-makers can plan the changes in policy direction in advance.

In this study, a system dynamics (SD) model is proposed to support efficient decision-making through policy effect analysis, from a macro-perspective for the performance improvement of school buildings. The SD model performs LCCA simulation based on performance improvement scenarios, to predict the deterioration pattern of school buildings and respond to it. Based on the simulation results, this study evaluates the long-term effects of rehabilitation policy on the performance grades of school buildings. Moreover, this study identifies an effective policy scenario that can achieve performance improvement.

## 2. Literature Review

Common methods of analyzing the complexity of asset management of social infrastructure include agent-based simulation (ABS) and SD. ABS is a micro-simulation method that can model interactions between agents; this method is used in various fields related to social infrastructures [8–12]. Echaveguren, Chamorro, and De Solminihac [13] modeled the interaction among agents (state, private and public) related to road infrastructure management systems, and analyzed the effects of the decisions made by agents regarding maintenance plans. Mallory, Crapper, and Holm [14] developed agent-based models (ABM) for fecal sludge (FS) recycling and proved the efficacy of the model by using case studies. Zechman [15] developed ABM for a water distribution system, and analyzed the interaction of systems. However, the ABS has a limitation in terms of modeling strategies, because the simulation results can differ based on small changes in the interaction method. Moreover, the level of detailed factors is high.

On the other hand, SD is a macro-simulation method that can decipher all the behaviors of complex systems [16]. In general, SD is used for modeling problems, such as performance measurement related to a social system, and estimating the effects of strategies and alternatives, as well as those of various social policies [17]. SD describes the interrelationships between factors causing changes in complex system growth estimates and patterns of change. SD emphasizes the causal relationships and feedback among individual components in a system [18]. Therefore, all causal relationships are recognized as circular relationships, without distinguishing between independent and dependent variables. This method focuses on the types of dynamic trends in changes among variables based on the flow of time, instead of obtaining the accurate value of the variable. Furthermore, SD is helpful when decision-makers examine the behavior of complex systems and evaluate the long-term policy effects [16]. Therefore, many studies have applied the SD modeling approach to determine the asset management strategy of social infrastructures in various fields.

Rehan [19] applied the SD modeling approach to develop asset management strategies for water and wastewater systems, and demonstrated the advantages of the SD model for modeling the interactions between physical, social, and financial systems. Mohammadifardi [20] also verified the applicability and efficacy of the SD model for wastewater collection (WWC). Hong, Frangmin, and Rongbei [21] developed an SD model related to highway maintenance issues, and proved that the SD method is effective during decision-making for a long-term plan by using case simulations and analyses. Soetjipto, Adi, and Anwar [22] developed a bridge deterioration model, and used it to simulate the possibility of bridge failure and detect that components that cause bridge failures. Furthermore, they used the SD model to analyze environmental pollution and energy problems, such as $CO_2$ emission in the transport industry [23,24]. Sing, Love, and Liu [25] proved that adopting the SD modeling approach is useful for dealing with the long-term rehabilitation policy of existing building stock, related to the sustainability of a city. Wang and Yuan [26] used an SD simulation to determine an optimal measure for effective risk management in infrastructure projects.

Therefore, various studies have shown that the SD modeling approach is an effective tool for exploring asset management strategies and the policy effects of social infrastructures. This approach is also used in various types of social infrastructures. However, studies are yet to use the SD modeling approach to investigate the performance improvement of school buildings. Therefore, this study proposes a model for the performance improvement of school buildings via the SD method.

## 3. Research Methodology

The overall study procedure is shown in Figure 1. A literature review is conducted to determine the conventional modeling methods of asset management, and to find a suitable model for this study. A decision-making model is then developed for performance improvement of the school building. The SD method is applied as a modeling method, and it is developed by considering the correlations among the deterioration, rehabilitation and finance models. The SD model used in this study is developed using the following sequence: (1) define the problem, (2) create a causal loop diagram (CLD), (3) create a stock and flow diagram (SFD), and (4) verify the model. For the completed SD model, the effectiveness of the decision-making model is proven by using case studies including data from school buildings. Moreover, suggestions for long-term planning and financing are provided for future performance improvement of school buildings, based on the test results of various policy scenarios.

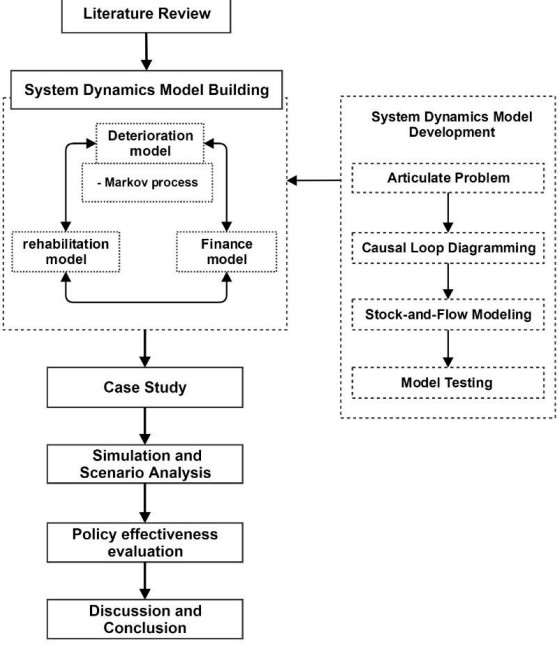

**Figure 1.** Research procedure.

## 4. Causal Loop Diagram of School Building Rehabilitation Management

During the first stage of SD model development, the causal relationships between key variables are determined to define the problem and compose the CLD for the school building rehabilitation system. A dynamic hypothesis is developed to explain the dynamic behavior of key variables in the structure. The overall system must be understood to establish a dynamic hypothesis, thereby emphasizing the need for conducting a literature review, expert group discussion, and survey. This study derived key variables and a dynamic hypothesis to understand the system by conducting a literature review. As a result of the relevant literature review, the SD model proposed in this study considers three major functions (asset deterioration, rehabilitation action, and total repair cost) for the macro-analysis of the rehabilitation system. Based on the literature review [7,19,27], nine variables, composing the three major functions, were derived. Figure 2 presents the CLD showing the causal relationships between the nine variables.

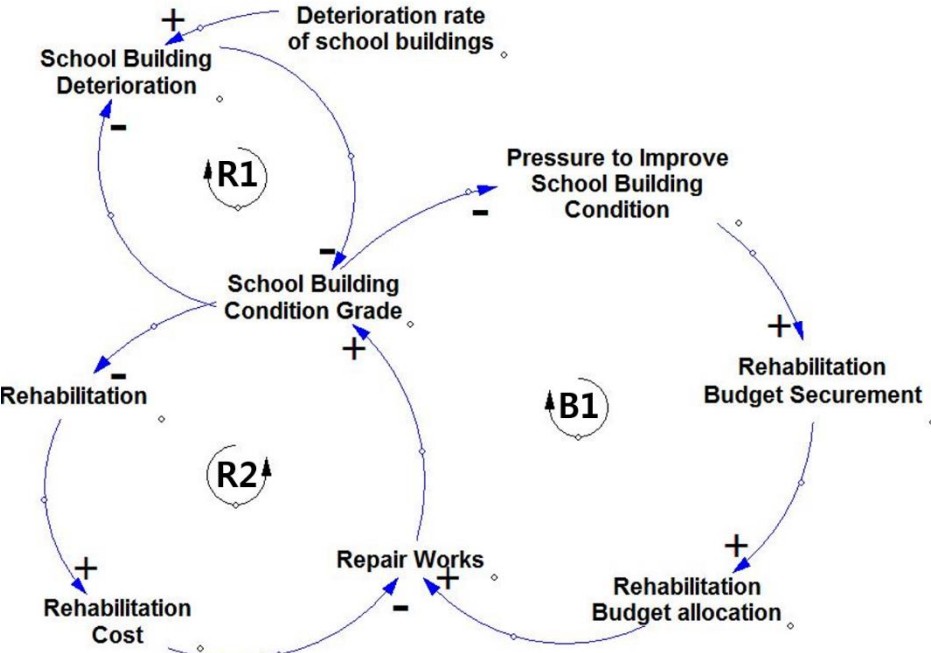

**Figure 2.** Causal loop diagram of the school building rehabilitation network management.

The CLD shown in Figure 2 consists of arrows, "+" or "−" signs, and feedback loops. A causal relationship between variables is expressed using the "+" or "−" signs through an arrow. These signs indicate the relationship between variables. The "+" link indicates that two variables (var1 and var2) are changing in a similar direction in the model. In other words, if the independent variable increases, the dependent variable also increases [Equation (1)].

$$\frac{\Delta \text{Var2}}{\Delta \text{Var1}} > 0 \tag{1}$$

The "−" link indicates that two variables (var1 and var2) are changing in different directions in the model. In other words, if the independent variable increases, the dependent variable decreases [Equation (2)].

$$\frac{\Delta \text{Var2}}{\Delta \text{Var1}} < 0 \tag{2}$$

The arrows of Figure 2 form the feedback loop, thereby indicating the characteristics of the loop. There are two types of loop, based on the characteristics of the loop: (1) reinforcing loop, and (2) balancing loop. The CLD presented in this study consists of two reinforcing loops (R1 and R2) and one

balancing loop (B1). Each feedback loop shows the dynamic behaviors of deterioration, rehabilitation, and rehabilitation finance (expenses and budget) for school buildings.

*4.1. Feedback Loop in School Building Deterioration*

The deterioration loop (R1) shows the representative physical deterioration process of social infrastructure. The two variables ("school building condition grade" and "school building deterioration") that form this loop are connected by the "−" link, thereby indicating that the "school building condition grade" negatively affects the "school building deterioration", and the "school building condition grade" is affected by the "school building deterioration". If the "school building condition grade" decreases (e.g., in the scale of A–E, whereby A is the optimal condition and E is the poor condition), the deterioration increases. If the deterioration increases, the "school building condition grade" decreases. Furthermore, the "deterioration rate of school building" is connected with the "school building deterioration" by the "+" link. Therefore, if the "deterioration rate of school building" increases, the "school building deterioration" increases. A combination of these links produces a reinforcing loop (R1), as depicted in Figure 2. A reinforcing loop includes the feature of reinforcing to the extreme of a certain side (thus causing an index growth/decrease behavior). Therefore, the deterioration loop (R1) establishes a cycle, wherein the condition deterioration of the school building accelerates as time elapses. For this dynamic behavior, a similar process has also been reported in many asset management studies and related references [7,19].

*4.2. Feedback Loop in Rehabilitation*

The rehabilitation loop (R2) shows the rehabilitation process of the school building. Because the deterioration loop (R1) causes the exponential deterioration of the school building, the "school building condition grade" decreases and the "rehabilitation action" increases. Therefore, the relationship between the two variables is connected with a "−" link. In the real world, monetary payments are required to perform maintenance and repair tasks during rehabilitation. Therefore, "rehabilitation cost" has a positive relationship ("+" link) with "rehabilitation action". If the "rehabilitation action" increases, the "rehabilitation cost" also increases. On the other hand, "rehabilitation cost" and "repair works" have a negative relationship ("−") link. This is because repair works can be performed only when sufficient rehabilitation budgets are supplied. Therefore, the rehabilitation loop (R2) shows the rehabilitation process of the school building, and the decrease of "repair works" indirectly indicates the decrease of "school building condition grade".

*4.3. Feedback Loop in Rehabilitation Finance*

The finance loop (B1) shows a budgeting process. If the "school building condition grade" decreases, users' condition improvement demands (those by students, teachers, staff, and local residents), and the managers of school facilities is increase. If the need for the school building's condition improvement is noted, the government can secure a budget for rehabilitation. The secured budget is appropriately allocated, based on the policies and plans. According to the final budget, the maintenance and repair tasks are performed, thereby improving the condition grade of the school building. This combination of links generates a balancing loop (B1), as described in Figure 2, and the finance loop (B1) mitigates the condition grade decline of the school building by the deterioration loop (R1) and the rehabilitation loop (R2).

Therefore, the "rehabilitation cost" of the feedback loop (R2), and the "rehabilitation budget allocation", directly affect the maintenance and repair tasks. Therefore, they affect the "school building condition grade".

## 5. Stock and Flow Modeling for System Dynamics Simulation

After understanding the overall feedback loop through the CLD, it should be converted into an SFD to perform computer simulations. System dynamics is a diagram-based programming language,

and the following diagrams constitute an SFD in the Vensim software: Stock, Flow, Valve, and Cloud (Figure 3a).

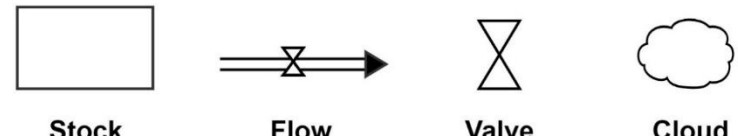

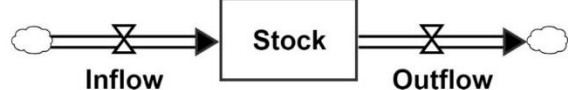

**Figure 3.** Stock and flow diagram.

Stock is a variable that accumulates or integrates the state of systems based on time. Flow is a variable that changes the value of the stock variable, and consists of inflow and outflow. Valve is a variable that controls the amount of inflow and outflow, and shows a boundary point of entry and exit of cloud. The relationship of stock and flow can be expressed using Equation (3), thus showing the value of the stock variable, based on the simulation time [16]. In this equation, $t_0$ is the initial time, t is the current time, and stock ($t_0$) is the initial value of stock. Inflow and outflow refer to flow coming into, and going out from, the stock, for an arbitrary duration (s) between the initial time ($t_0$) and the current time (t). Equation (4) determines changes in the rate of stock, based on time [16].

$$\mathrm{Stock(t)} = \int_{t_0}^{t} [\mathrm{Inflow(s)} - \mathrm{Outflow(s)}]ds + \mathrm{Stock(t_0)} \tag{3}$$

$$\frac{d(\mathrm{Stock})}{dt} = \mathrm{Inflow(t)} - \mathrm{Outflow(t)} \tag{4}$$

The relationship of stock and flow can be expressed according to the aforementioned, as shown in Figure 3b.

### 5.1. School Building Deterioration Sector

The school building deterioration model in this study is developed with the goal of simulating the overall deterioration pattern. Most assets are managed based on the school building condition; deterioration models using data regarding the condition have been presented using various methods. Based on the results of the literature review, deterioration models are primarily classified into three categories: deterministic, stochastic, and artificial intelligence [28,29] (Figure 4).

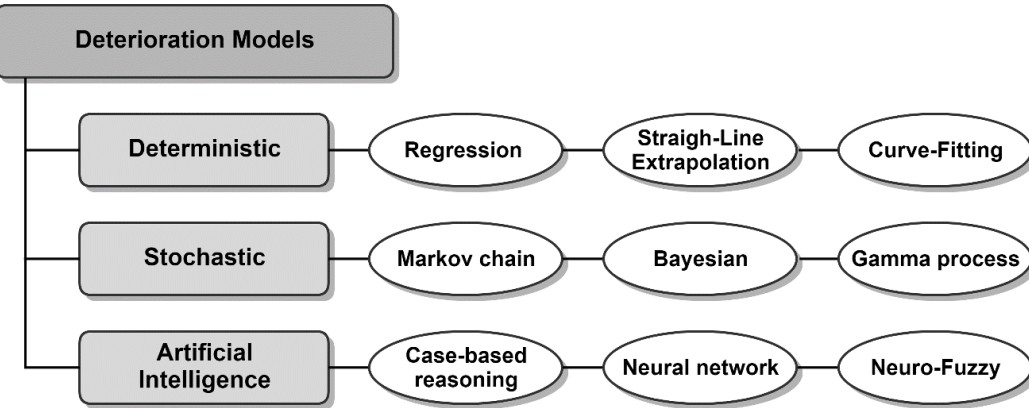

**Figure 4.** Classification of deterioration models.

Among the categories of deterioration models, Markov chain is a stochastic method for predicting the future condition state of assets in a social infrastructure management system. This method is also most frequently used [30–33]. Therefore, this study attempts to predict the deterioration pattern of school buildings by using the Markov chain.

The Markov chain indicates a case wherein the probability of reaching a specific state for a stochastic variable depends only on the state of the preceding time point. This study classifies the physical condition of school buildings using grades A–E, according to the condition evaluation criteria provided by the Ministry of Education in South Korea (Table 1).

**Table 1.** Physical condition grade of school buildings.

| Grade | Grade Point | Description |
|-------|-------------|-------------|
| A | 5 | Exceptional: Fit for the future |
| B | 4 | Good: Adequate for now |
| C | 3 | Mediocre: Requires attention |
| D | 4 | Poor: At risk |
| E | 1 | Critical: Unfit for the future |

The deterioration model is developed based on the assumption that a school building deteriorates to the next condition state only from a specific condition state (e.g., from condition A to B, and B to C). The five stock variables (A–E) shown in Figure 5 indicate the number of school buildings for each condition grade. A transition probability variable is derived, based on case study data, that serves as an auxiliary variable of each flow variable. Moreover, to induce a pattern that is similar to the actual deterioration behavior of assets, the stock variable has a feedback relationship that affects the flow variable. This variable can be expressed as shown in Equation (5) (X indicates the condition grade, and X-1 refers to the condition grade that is one step lower than that of the condition X).

$$\text{Deterioration X to X} - 1 = \text{X} * \text{Transition Probability X to X} - 1 \qquad (5)$$

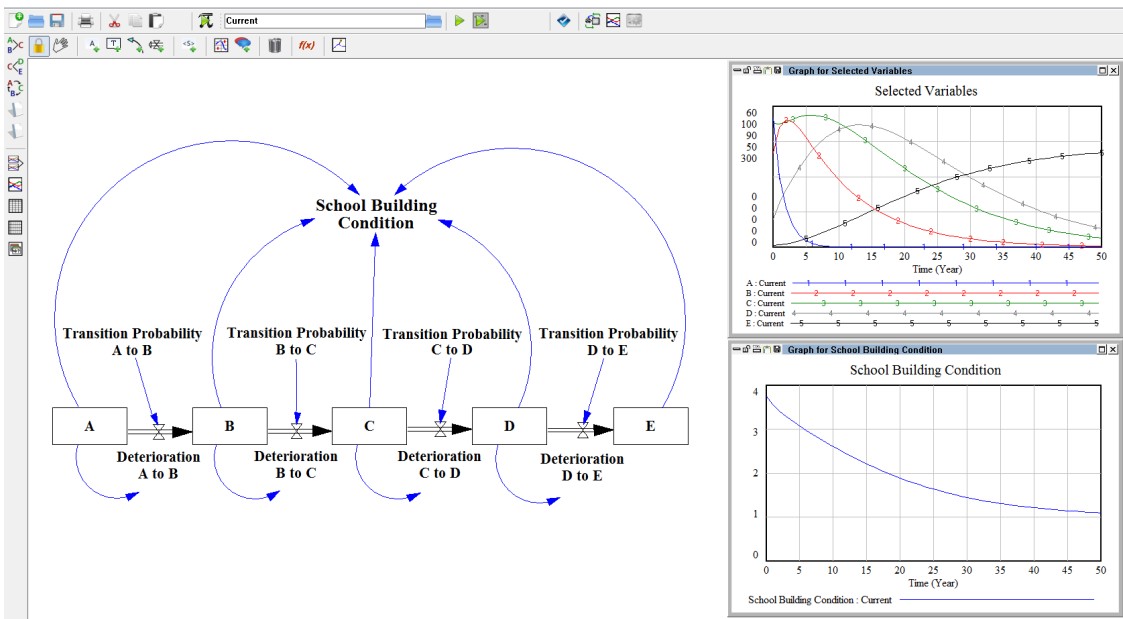

**Figure 5.** System dynamics (SD) model of deterioration and simulation graph.

Moreover, to identify the condition grade of overall school buildings based on time, Equation (6) was applied, based on the grade score shown in Table 1.

$$\text{School building condition} = (\text{A} * 5 + \text{B} * 4 + \text{C} * 3 + \text{D} * 2 + \text{E} * 1) \;/\; \text{Total number of school building} \tag{6}$$

To test the completed deterioration model, data regarding the safety inspection and condition assessment of school buildings for the winters of 2014–2018, from the Education Office in Daegu metropolitan city, was used in this study. The condition grades of 214 school buildings in total showed: grade A = 55, grade B = 67, grade C = 79, grade D = 10, and grade E = 3 buildings. The transition probability was set as: A to B = 0.45, B to C = 0.1, C to D = 0.09, and D to E = 0.15 (the transition probability of school buildings that are applied in the case studies are described in detail in Section 5.1). The result of testing the deterioration model using the case study data is shown in the graph on the right side of Figure 5.

The result of testing the deterioration model using the case study data is shown in the graph on the right side of Figure 5. As time elapses, the number of school buildings with the condition grades A, B, C, and D decreases, and the number of school buildings with the condition grade E increases. The curve illustrating the comprehensive condition of school buildings based on these dynamic changes has an initial value of 3.75, which is close to the grade B. However, after 50 years, the value deteriorates to 1.09, thus the school building condition grade deteriorates to grade E. Therefore, this study verified the validity of the deterioration SD model as a tool that predicts deterioration patterns, using the number of assets by grade.

### 5.2. Rehabilitation Sector

The rehabilitation model shows the rehabilitation action based on the condition grades of school buildings. The model proposed in this study assumes that schools categorized under the three grades, C, D, and E, which do not indicate a good condition, will be repaired to ensure the school building is categorized under the best grade A. Based on this assumption, the rehabilitation action was integrated with the deterioration models shown in Figure 6.

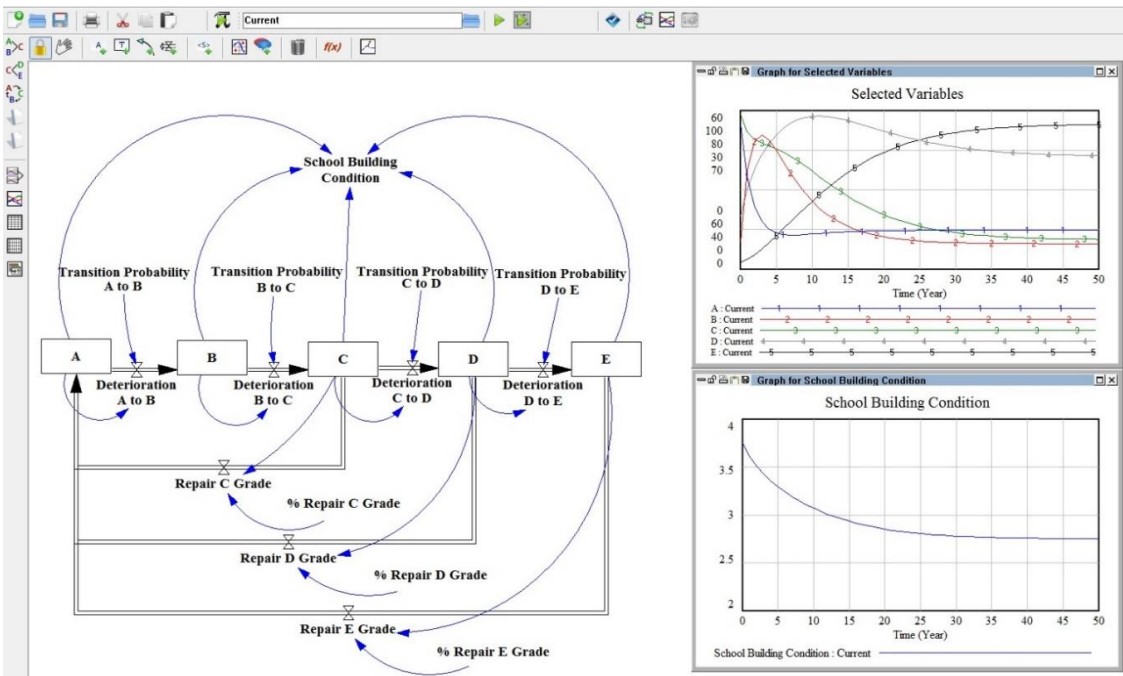

**Figure 6.** SD model of rehabilitation and simulation graph.

The dynamic flow of flow variables (e.g., Repair C Grade), that expresses the rehabilitation action in Figure 6, is pointed toward an improved condition state (grade A) from a specific condition state (grades C, D, or E). The value of the rehabilitation flow variable is determined based on an auxiliary variable (e.g., % Repair C Grade). This variable shows the proportion of repairing from condition grade X to grade A in the number of school buildings of a specific condition grade. The flow variable showing the rehabilitation action of the SD model is calculated by Equation (7).

$$\text{Repair X Grade} \; = \; \text{X} \; * \; \% \text{ Repair X Grade} \tag{7}$$

Equation (7) determines the number of buildings of each condition X (grades C, D, or E). Stock A—the number of school buildings that secured grade A—increases through the rehabilitation action. This is expressed using Equation (8)

$$\begin{aligned} \text{Stock A(t)} \; = \int_{t_0}^{t} [&\text{Repair C Grade(s)} + \text{ Repair D Grade(s)} + \text{Repair E Grade(s)} \\ &-\text{Deterioration A to B(s)}]\text{ds} + \text{Stock A(t}_0) \end{aligned} \tag{8}$$

Equation (8) indicates the inflow into Stock A, which refers to the number of school buildings that have been improved from the grades C, D, and E, during an arbitrary time period between the initial time $t_0$ and the current time t. Equation (8) also indicates the outflow to Stock B caused by deterioration as time elapses.

The auxiliary variables—% Repair X Grade—were set to 5% to test the model that included rehabilitation action. For a case concerning the performing of rehabilitation actions to the school buildings with grades C, D, and E every year, the result of simulating the condition grade changes over 50 years is shown in a graph on the right side of Figure 6. A comparison of the number of school buildings in each grade, described on the right sides of Figures 5 and 6, indicates that the graph curves of grades A, B, C, and D in Figure 5 decrease rapidly, whereas the grades A, B, C, and D in Figure 6 maintain specific levels for approximately 25 years.

Table 2 shows the simulation results, on a yearly basis, for the condition grade values of all school buildings for 50 years, targeting the deterioration (Det.) and rehabilitation (Reh.) models. Based on the 50 year period, the condition grade value in the deterioration model was 1.09, which was critical.

However, the value improved to 2.75 in the rehabilitation model, thus indicating a poor condition. The results of repairing 5% of the school buildings in grades C, D, and E to grade A, respectively, every year are shown in Table 2.

**Table 2.** Comparison of simulation results of school building physical condition grade between deterioration model and rehabilitation model.

| Model | Simulation Time (Year) | | | | | | State |
|---|---|---|---|---|---|---|---|
| | 0 | 5 | 10 | 15 | 25 | 50 | |
| Det. | 3.75 | 3.07 | 2.60 | 2.21 | 1.63 | 1.09 | critical |
| Reh. | 3.75 | 3.29 | 3.07 | 2.93 | 2.80 | 2.75 | poor |

Moreover, Table 3 compares the differences in the deterioration model (Det.) and rehabilitation model (Reh.), based on their respective grades for the same simulation results. Although the initial value (0 year) was identical, the number of school buildings for each grade indicated a significant difference between the two models as time elapsed. Based on the 50 year period, the Det. model showed that most school buildings deteriorated to grade E, whereas the number of school buildings was evenly distributed in the Reh. model. Therefore, it is proven that the rehabilitation SD model in Figure 6 can quantitatively analyze the effects of school buildings' deterioration and rehabilitation action on the increase or decrease of the physical conditions of all school buildings.

**Table 3.** Comparison of simulation results between deterioration model and rehabilitation model.

| Grade | State | 0 Year | | 5 Year | | 10 Year | | 15 Year | | 25 Year | | 50 Year | |
|---|---|---|---|---|---|---|---|---|---|---|---|---|---|
| | | Det. | Reh. | Det. | Reh. | Det. | Reh. | Det. | Reh. | Det. | Reh. | Det. | Reh. |
| A | Exceptional | 55 | 55 | 2.76 | 12.9 | 0.13 | 13.47 | 0.007 | 14.21 | 0 | 14.67 | 0 | 14.75 |
| B | Good | 67 | 67 | 77.76 | 87.1 | 47.83 | 75.46 | 28.3 | 70.7 | 9.88 | 66.99 | 0.7 | 66.42 |
| C | Mediocre | 79 | 79 | 84.22 | 69.5 | 79.26 | 63.39 | 65.5 | 57.28 | 37.07 | 50.43 | 5.7 | 47.54 |
| D | Poor | 10 | 10 | 31.45 | 26.6 | 41.69 | 28.90 | 42.9 | 27.71 | 31.79 | 24.05 | 6.6 | 21.48 |
| E | Critical | 3 | 3 | 17.79 | 17.7 | 45.06 | 32.75 | 77.1 | 44.72 | 135.2 | 57.04 | 200 | 63.78 |

*5.3. Finance Sector*

One of the crucial tasks in a maintenance and rehabilitation (M&R) plan is efficiently distributing a limited fund to achieve an optimal outcome. This section aims to propose an SD model that has added a cost model to the deterioration and rehabilitation models for efficient budget allocation.

Figure 7 is an integrated SFD, whereby the cost model is included in Figure 6. To fulfill the rehabilitation action of a deteriorated school building, maintenance costs are required, which are provided from a limited budget. The "Available Rehabilitation Policy Budget" variable refers to the total budget that can be used for the rehabilitation action in the model. The "Allocated Budget to Repair X Grade" variable refers to a budget allocated from the limited total budget to repair the school buildings belonging to the respective grades X (C, D, and E) [Equation (9)].

$$\text{Allocated Budget to Repair X Grade} = \text{ Available Rehabilitation Policy Budget} * \text{ "\% Budget to Repair X Grade"} \tag{9}$$

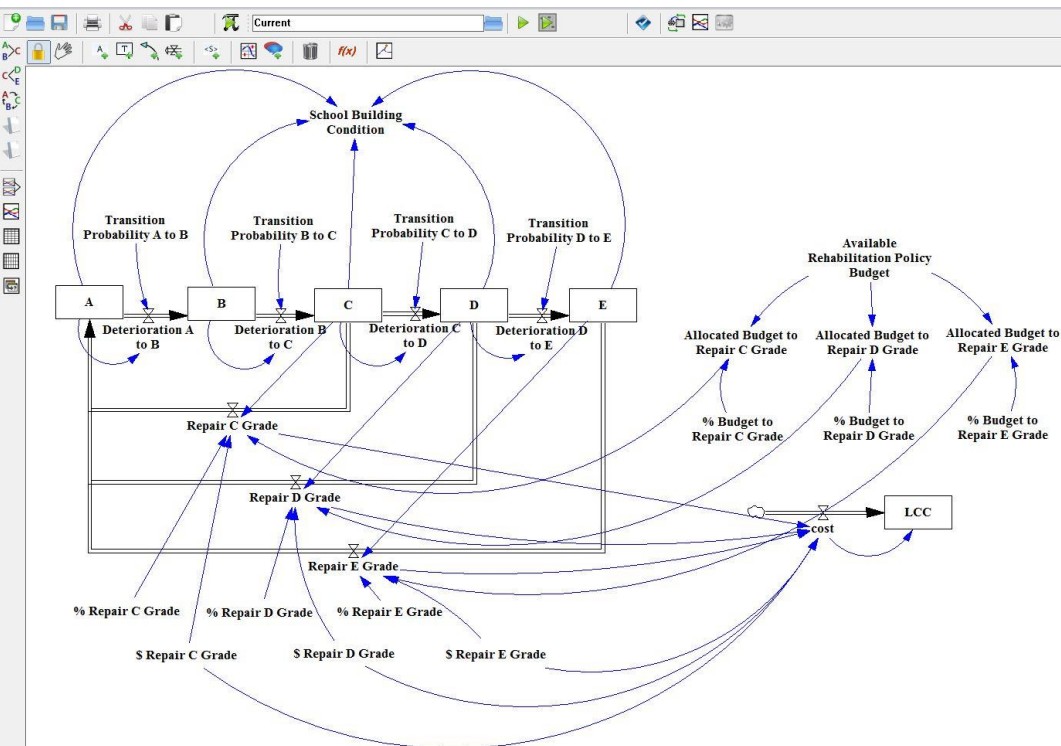

**Figure 7.** SD model for rehabilitation cost and budgeting analysis.

The value of this variable is determined by a variable "% Budget to Repair X Grade", which shows the percentage of budget allocated to each grade X. Moreover, unlike the rehabilitation model, the cost model shows that the number of school buildings restored is limited according to the allocated budget. This is determined through Equation (10).

$$\begin{aligned}
\text{Repair X Grad} = \text{IF THEN ELSE}(&\text{X} * \text{\% Repair X Grade} * \text{\$ Repair X Grade} \\
&< \text{ Allocated Budget to Repair X Grade},\ \text{X} * \text{\% Repair X Grade}, \\
&\text{X} * \text{\% Repair X Grade} * 0)
\end{aligned} \tag{10}$$

Equation (10) used the IF THEN ELSE({cond}, {ontrue}, {onfalse}) function, which is a built-in function of Vensim, to derive different values, based on the condition. The variable "$ Repair X Grade" shows the repair cost required to rehabilitate the school buildings of each grade. If the cost for repairing from a condition grade X (C, D, or E) to the grade A (X * "% Repair X Grade" * "$ Repair X Grade") is less than the limited budget ("Allocated Budget to Repair X Grade"), the rehabilitation action is conducted; otherwise, it is stopped. Moreover, the life cycle cost by time t shows an equation similar to Equation (11).

$$\begin{aligned}
\text{LCC(t)} \quad = \int_{t_0}^{t} [\text{\$ Repair C Grade} &* \text{Repair C Grade(s)} + \text{\$ Repair D Grade} \\
&* \text{Repair D Grade(s)} + \text{\$ Repair E Grade} * \text{Repair E Grade(s)}] ds
\end{aligned} \tag{11}$$

The completed integrated SD model is used as a model for determining the life cycle cost analysis and future outcome prediction, to improve the performance of school buildings via use of a case study. This study performs a scenario analysis, to investigate the effects of budget allocation by grade on the total outcome and cost, based on simulations considering various values for the "% Budget to Repair X Grade" variable.

## 6. Application of the Developed SD Model

This section performs the budget allocation scenario analysis, to predict the deterioration behavior of school buildings and improve performance by simulating the SD model proposed in Section 5. This study used the safety inspection and condition assessment data provided by the Ministry of Education in South Korea, in investigating 214 school buildings in a metropolitan city, Daegu. Based this data, this study acquired data regarding the 5 year (2014–2018) condition assessment and maintenance cost of school buildings, categorized by condition grade. At present, the Ministry of Education in South Korea designates only grades D and E, among the five condition grades A–E, as disaster-prone buildings, and conducts performance improvement primarily for these buildings. However, the SD simulation sets the rehabilitation scenarios by considering buildings up to the grade C for preventive maintenance. Finally, the simulation analysis is performed by applying the deterioration rate variable (transition probability matrix, TPM), derived by using the Markov chain stochastic process, to the integrated SD model (Figure 7).

*Markov Approach*

Markov chain is a discrete time stochastic process, and the conditional probability of a specific future event changes according to only the current condition; it is unrelated to past conditions [34]. Because five condition states exist in the case study data, the transition probability from one condition state to another is expressed in a $5 \times 5$ matrix, and the simplified transition probability matrix (TPM) is shown in Equation (12).

$$\text{TPM} = \begin{bmatrix} 0.88 & 0.12 & 0 & 0 & 0 \\ 0 & 0.96 & 0.04 & 0 & 0 \\ 0 & 0 & 0.91 & 0.09 & 0 \\ 0 & 0 & 0 & 0.86 & 0.14 \\ 0 & 0 & 0 & 0 & 1 \end{bmatrix} \tag{12}$$

Each element ($P_{ij}$) of TPM shows a probability (P) of transitioning from a state "i" to another state "j". For example, '0.88' indicates the probability of transitioning from state A to state A (the probability of a school building remaining in state A), and '0.12' refers to the probability of transitioning from state A to state B. It is assumed that the condition state of school buildings shift from one condition state to the next condition state only. Suppose the initial condition state's value is CS0, and the distribution of the condition state by year is n; then, Equation (13) can be derived.

$$\text{CS}_n = \text{CS}_0 \times \text{TPM}^n \tag{13}$$

Equation (13) shows that a future state (CS0) can be estimated when the TPM and the initial state (CSn) are known.

## 7. SD Model Simulation Results of Scenario Analysis

The developed SD model (Figure 7) analyzes the effect of budget allocation strategy scenarios to determine cost-effective rehabilitation actions for school buildings. Table 4 shows the budget allocation proportions of grades C, D, and E, based on the average annual educational environment improvement budget provided for Daegu city. The results of simulating 10 scenarios using the Vensim software are shown in Figure 8.

**Table 4.** Budget allocation scenarios.

| Scenario | Budget to Repair C Grade | Budget to Repair D Grade | Budget to Repair E Grade |
|---|---|---|---|
| S1 | 0 % | 0 % | 100% |
| S2 | 0% | 100% | 0% |
| S3 | 100% | 0% | 0% |
| S4 | 33% | 33% | 34% |
| S5 | 0% | 50% | 50% |
| S6 | 50% | 50% | 0% |
| S7 | 50% | 0% | 50% |
| S8 | 25% | 25% | 50% |
| S9 | 25% | 50% | 25% |
| S10 | 50% | 25% | 25% |

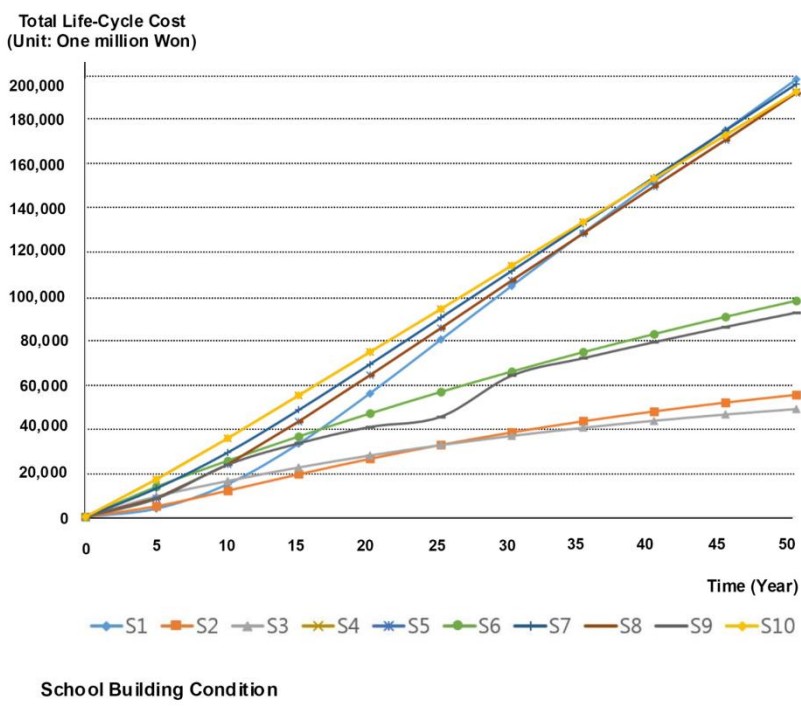

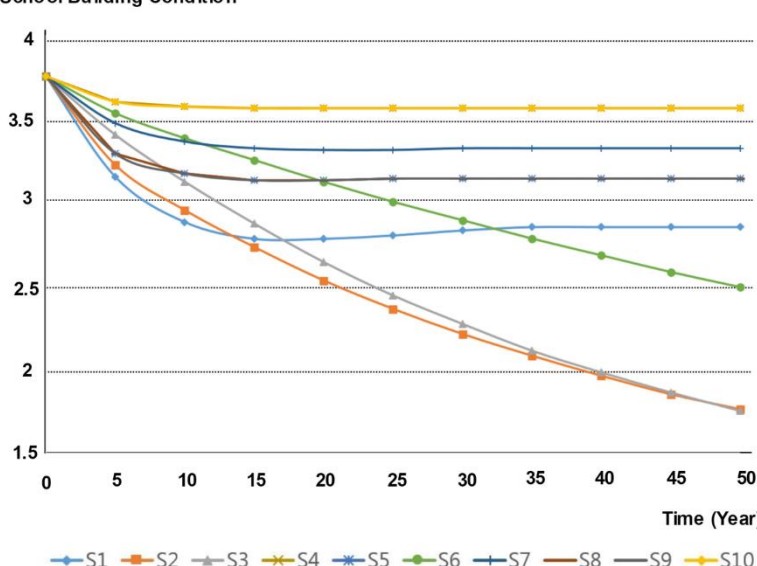

**Figure 8.** SD model simulation analysis results for budget allocation scenarios.

Figure 8 shows the scenario analysis results for the condition grades (based on Table 1) and Total life cycle cost (TLCC) of all school buildings. S2 and S3 can reduce the TLCC in the long term. However, because the condition grade of school buildings declines gradually to 1.75 (grade E: Critical), they can be perceived as the worst-case scenarios. TLCC of approximately KRW 20 billion is expected for S1, S4, S5, S7, S8, and S10, and TLCC of approximately KRW 9 billion is expected for S6 and S9. Therefore, S9, S10, and S4 are picked as scenarios with good performance improvement effects relative to the cost. From these results, it can be noted that the appropriate budgets were allocated to school buildings in the condition grade C. In the case of S10, wherein the condition grade was the highest, 50% of the total budget was allocated to the condition grade C and the remaining 50% was equally allocated to the condition grades D and E. This result shows that, when repairing is performed primarily on buildings in condition grade C, the rehabilitation cost required is less than that of buildings in grades D and E, and in the long term, a preventive maintenance effect can be obtained. Therefore, by using the 10 different scenario analyses, it is ascertained that budget allocation based on condition grade has a crucial impact on the total school building performance.

## 8. Discussion and Conclusions

This study proposed an integrated SD model for the rehabilitation policy analysis of school buildings. To validate the SD model, 10 rehabilitation budget allocation scenarios were analyzed, based on the simulations. The results show that the integrated SD model can support strategic decision-making, by identifying the school building condition grades and TLCC behavior for each scenario in the long-term perspective. According to the scenario analysis, the rehabilitation action of preventive maintenance that primarily repairs the buildings in condition grade C showed the best performance improvement effect relative to the cost.

The Ministry of Education in South Korea currently performs post-event maintenance management, to repair buildings when performance deterioration occurs (grades D and E). However, the preventive maintenance method should be adopted to reduce the deterioration speed of school buildings. The costs calculated based on the SD simulation can be used for the long-term planning of rehabilitation action, by estimating the cost that will be injected into repairing the deteriorated school buildings for 50 years in the future. However, the proposed SD model has several limitations. The available case study data for this study was insufficient, and increasingly accurate deterioration modeling will be possible if it is supplemented with an optimal method for estimating accurate TPM with limited data. Moreover, the budget of the Ministry of Education in South Korea, which is the subject of the case study, is in a situation wherein continuous investment for the performance improvement of school buildings is difficult because of other educational policies, such as free school meals and the New University for Regional Innovation (NURI) project. Therefore, if the proposed SD model is expanded to consider the effects of other educational policies, the crucial performance improvement budget can be estimated in the long-term perspective.

**Author Contributions:** Conceptualization, S.K. (Suhyun Kang), S.K. (Sangyong Kim), S.K. (Seungho Kim), and D.L.; data curation, D.L.; formal analysis and investigation, S.K. (Suhyun Kang), S.K. (Sangyong Kim); methodology, S.K. (Suhyun Kang), S.K. (Sangyong Kim), S.K. (Seungho Kim); resources, D.L.; software, S.K. (Suhyun Kang) and S.K. (Seungho Kim); supervision, S.K. (Sangyong Kim) and D.L.; validation, S.K. (Suhyun Kang), S.K. (Sangyong Kim), S.K. (Seungho Kim), and D.L.; visualization, S.K. (Suhyun Kang); writing—original draft, S.K. (Suhyun Kang), S.K. (Seungho Kim); writing—review and editing, S.K. (Sangyong Kim) and D.L. All authors have read and agreed to the published version of the manuscript.

**Funding:** This work was supported by the National Research Foundation of Korea (NRF) grant funded by the Korea government (MSIT) (No. NRF-2018R1A5A1025137).

**Conflicts of Interest:** The authors declare no conflict of interest.

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
