# Peer review of "System Dynamics Model for the Improvement Planning of School Building Conditions"

_sustainability, doi:10.3390/su12104235_

Round 1

Reviewer 1 Report

The topic of long-term management strategies for buildings faced by the authors addresses the current issue of sustainable performance of buildings.

It is clear that the study has the purpose of further investigating this important issue, that has also an impact on people safety, with the goal to improve the response to building deterioration by proposing a preventive maintenance method rather than a breakdown method that is also financially sustainable for the governments.

The paper structure is well organized. The study setup is efficiently presented, therefore delivering a clear understanding of both the aim of the study and the methods used to carry out the experiments.

Therefore I require minor improvements before submitting the paper for publication.

My general comments are:

  1. Lines 45 – 52: the Introduction section addresses past studies, therefore overlapping with the following Literature review section;
  2. The research methodology is effective and well explained however, the modeling methods section is significantly longer than the proposed model application section. In my opinion the application of the developed model should be more widely argued.

My specific comments are:

  1. Lines 422 – 423: the repetition of the same sentence twice is a typo;

Author Response

Reviewer # 1.

1. Lines 45 – 52: the Introduction section addresses past studies, therefore overlapping with the following Literature review section;

Answer:

Thank you for your suggestion. We have reviewed the Introduction and deleted sentences that overlap with the Literature Review.

2. The research methodology is effective and well explained however, the modeling methods section is significantly longer than the proposed model application section. In my opinion the application of the developed model should be more widely argued.

Answer:

SD methodologies simulate dynamic hypotheses between variables using computers for complex system analysis. Therefore, for the validity and reliability of the proposed SD model, we decided that setting up a correlation that expresses the causal relationship between variables qualitatively and quantitatively is important. Consequently, this paper focused on SD modeling methods to analyze interactions between deterioration, rehabilitative activities, and finance models for school buildings over time. I wanted to include everything in this paper, but I thought it would be better to divide it into theoretical studies and case study papers. This paper corresponds to the theoretical study, and a detailed study of the proposed model application is being conducted to confirm the actual applicability in the trailing study.

3. Lines 422 – 423: the repetition of the same sentence twice is a typo;

Answer:

Thank you for your suggestion. We have deleted repetitive sentences.

Again, thank you for giving us the opportunity to strengthen our manuscript with your valuable comments and queries. We have worked hard to incorporate your feedback and hope that these revisions persuade you to accept our submission.

Reviewer 2 Report

This is quite an interesting paper that proposes a novel system dynamics model for evaluating evaluating different investment and rehabilitation strategies for schools. the model structure and operational systems would seem to be applicable to other contexts.  The authors have highlighted the limitations of the current model and suggested how future work could enhance the accuracy and relevance of the model for use.   

Author Response

Answer:

Thank you for your favor. We will overcome the limitations of the model presented in the trailing study and improve the accuracy of the model and relevance of the model for use.

Again, thank you for giving us the opportunity to strengthen our manuscript with your valuable comments and queries. We have worked hard to incorporate your feedback and hope that these revisions persuade you to accept our submission.